# Admission to the Emergency Department by Patients Being Followed up for Palliative Care Consultations

**DOI:** 10.3390/ijerph192215204

**Published:** 2022-11-17

**Authors:** Mariana Azevedo Brites, Joana Gonçalves, Francisca Rego

**Affiliations:** 1Faculty of Medicine, University of Porto, 4200-319 Porto, Portugal; 2Family Health Unit Corino de Andrade, Póvoa de Varzim, 4490-602 Póvoa de Varzim, Portugal; 3Póvoa de Varzim—Vila do Conde Hospital Center, 4490-421 Póvoa de Varzim, Portugal

**Keywords:** emergency service, hospital, palliative care, health services misuse

## Abstract

Introduction: Palliative care aims to improve the quality of life of patients and families facing life-threatening diseases. Admissions to the emergency department are considered potentially avoidable. This study aims to characterize the use of the emergency department by palliative care patients at a public hospital in Portugal. Methods: This retrospective study included patients who had their first palliative care appointment during the year 2019; 135 patients were included, with 255 admissions to the emergency department. Descriptive statistical analysis consisted of calculating the absolute (*n*) and relative (%) frequencies for categorical variables and medians (Mdn) and percentiles (P25 and P75) for continuous variables. The multivariable associations were calculated via logistic models, with the statistical significance set to *p* < 0.05 and 95% confidence intervals. Results: Dying in hospital was associated with going to the emergency department. Patients who died in hospital had more admissions and spent more time there. Conclusion: Emergency department admissions suggest that there are gaps in the provision of care. It is necessary to anticipate crisis situations, provide home and telephone appointments, and invest in professionals’ education to respond to the needs that will grow in the future.

## 1. Introduction

Cicely Saunder founded the St. Christopher’s Hospice in 1967, laying the foundations of the current palliative care movement and the need to implement end-of-life care. The initial use of the word “palliative” dates back to 1975 when the first palliative care service at the Royal Victoria Hospital in Montreal was created [1]. The World Health Organization has defined palliative care as an approach to improve the quality of life of patients and families who are facing problems associated with life-threatening conditions. Its mission is to prevent and alleviate suffering through the early identification and treatment of pain and other problems, whether physical, psychosocial, or spiritual [2,3].

More than 56.8 million individuals are estimated to be in need of palliative care globally, including 31.1 million before and 25.7 million near the end of life. [4]. It was used in 45.3% of all deaths in 2017 [4]. Oncological disease accounts for about 28% of cases followed by human immunodeficiency virus (22%), cerebrovascular disease (14%), and dementia (12%) [4]. Palliative patients from all over the world, aged over 20 years old, experienced 20 million days of suffering due to poor control of symptoms, namely, fatigue, pain, depression, anxiety, dyspnea, confusion, and delirium [3]. These symptoms lead to admissions to the emergency department (ED) and hospital admissions, which, along with hospital death, negatively interfere with quality of life [5].

Studies show that potentially avoidable admissions to the emergency department—defined as accesses to the emergency department that could have been managed in another type of health service or even avoided with preventive care—accounted for 19 to 58% of episodes [6,7,8,9]. These are associated with poor disease management, inadequate availability of services, and lack of patient literacy for decision-making [10]. Decreased efficiency and increased healthcare costs are consequences of avoidable emergency department visits [10]. Palliative care is cost effective and simple to apply at home; patients prefer it [11,12].

It is estimated that 400,000 health professionals are involved in the provision of palliative care, which represents only 0.9% of the total worldwide [13]. The European Association of Palliative Care recommends two specialized palliative care services per 100,000 inhabitants; i.e., a hospital care team and a home care team [13]. The growing need for palliative care implies that health systems know the reality, so that they are prepared to respond to it both in human and organizational terms [13].

This study aimed to characterize the use of the emergency department by patients followed up in palliative care consultations between time of admission and time of death. Firstly, we wanted to characterize and identify differences between users and non-users of the emergency department. Secondly, we intended to characterize patients and the episodes in the emergency department and identify the factors associated with the use of the emergency department.

## 2. Materials and Methods

This retrospective study included patients who had their first consultation in palliative care at a public hospital in northern Portugal in 2019. We considered an average of 207 per year and the sample size was calculated taking into account a confidence level of 95% and a margin error of 5%, with random sampling, and obtained the necessary sample of 135 patients. [14]. The study included deceased adults aged 18 years or more for whom there was complete information in their clinical files. Patients who were alive at the time of data collection, in May 2022, were excluded (*n* = 10). A total of 135 patients were included in the study and accounted for 65.2% of the total patients admitted for palliative care consultation in 2019. This led to 255 admissions to the emergency department. The study was authorized by the Ethics and Data Protection Committee of the Hospital Center. The hospital center where the study took place has a palliative care team whose mission is to support patients and families in need of differentiated palliative care. It is a multidisciplinary team that includes doctors, nurses, a psychologist, and a social worker, working Monday to Friday, 8 a.m. to 5 p.m. Referral to the consultation can be carried out by doctors of any of the hospital services or by family doctors. The team has a telephone contact, available Monday to Friday, from 8 a.m. to 5 p.m., for patients and families as well as requests for advice from other health professionals. It is a consulting team without its own inpatient service and does not provide home care services. Home palliative care service is not available in this region.

Data were collected to characterize the patients and their use of health services: gender, age, marital status, residence (domicile, nursing home, continuing care unit, host family), time between admission to palliative care consultation and death, number of consultations between admission to palliative care consultation and death, number of hospitalizations between admission to palliative care consultation and death, time in the emergency department, place of death, death in the last admission to the emergency department or subsequent hospitalization, and hospitalization proposed in palliative care consultation. It also included data to characterize the episodes of admission to the emergency department: main complaint, number of complaints, time of admission, origin, request for the PC team’s opinion, length, and diagnosis. Statistical analysis was performed using SPSS, version 26.0. For description of the variables, the absolute (*n*) and relative (%) frequencies were used for categorical variables and medians (Mdn) and percentiles (P25 and P75) for the continuous variables, given their asymmetric distribution. The univariate associations of the categorical variables were evaluated using the chi-squared test or Fisher’s test in case the former’s assumptions were not met. Univariate associations of the continuous variables were evaluated using the Mann–Whitney test or the Kruskal–Wallis test for two (or more than two) groups. Multivariable associations were evaluated by constructing logistic models in case the dependent variables are binary. The work used a respective calculation of the adjusted odds ratio (aOR) and by constructing Poisson models in cases where the dependent variables are counts with a right-skewed distribution, with calculation of the adjusted relative risk (aRR). The inclusion of independent variables in these models was based on the criterion *p* < 0.10 in univariate analyses, to adjust variables that are more comprehensive. This, in turn, increases the models’ explanatory power. In multivariable models, the decision of statistical significance was guided by the criterion *p* < 0.05, with the confidence intervals, for this purpose, set at 95%.

## 3. Results

### 3.1. Characterization of patients

There were 135 patients. Most were male (*n* = 75, 55.6%) and married (*n* = 87, 64.4%). Oncological disease was the most frequent admission diagnosis to palliative care consultation (*n* = 87, 64.4%), followed by other causes (cardiovascular and respiratory disease) (*n* = 25, 18.5%) and dementia syndrome (*n* = 23, 10.0%). Most patients lived at home (*n* = 117, 86.7%). The most prevalent place of death was the hospital (*n* = 66.48.9%). The median age was 82.0 years (P25 = 67.0, P75 = 88.0). The median follow-up time in consultation was two months (P25 = 1.0, P75 = 5.0). The median number of consultations was five (P25 = 2.0, P75 = 10.0) (Table 1).

Most patients (*n* = 98, 72.6%) went to the Emergency Department. The median length per episode of the emergency department was five hours (P25 = 3.0, P75 = 9.0). The median time interval between the last episode in the emergency department and death was five days (P25 = 1.0, P75 = 15.0). Most deaths occurred during the last visit to the emergency department (*n* = 56, 57.1%).

### 3.2. Differences and Associations Regarding Patients Admitted to the Emergency Department

When including the variables that showed statistically significant differences in a logistic model, the results suggest that having at least one hospitalization (aOR = 6.65, 95%CI = 1.84, 24.07) and dying in hospital (aOR = 9.20, 95%CI = 2.42, 35.07) are associated with resorting to the emergency department (Table 2).

In the multivariable Poisson model, the variables that showed association in the univariate analysis were included, even those that showed marginal significance (*p* < 0.10). More than one hospitalization (aRR = 2.26, CI95% = 1.52–3.37), death during the last visit to the emergency department (aRR = 1.48, CI95% = 1.09–2.00) and longer follow-up time in palliative care consultations (aRR = 1.03, CI95% = 1.01–1.06) were associated with an increased risk of more episodes in the emergency department (Table 3).

In the multivariable Poisson model, the variables that showed association in the univariate analysis were included, even those that showed marginal significance (*p* < 0.10). Not having a partner (aRR = 1.56, 95%CI = 1.40–1.73), one hospitalization (aRR = 1.62, 95%CI = 1.39–1.89), more than one hospitalization (aRR = 2.80, 95%CI = 2.37–3.32), death during the last visit to the emergency department (aRR = 1.78, 95%CI = 1.57–2.01), longer follow-up time in palliative care consultations (aRR = 1.05, 95%CI = 1.04–1.07), and greater number of face-to-face or telephone consultations (aRR = 1.02, 95%CI = 1.01–1.03) were associated with an increased risk of longer total length of episodes in the emergency department (Table 4).

### 3.3. Characterization of Admissions to the Emergency Department

There were 251 emergency episodes. More than half (*n* = 126, 50.2%) were due to respiratory complaints followed by pain (*n* = 34, 13.5%), changes in behavior and general status (*n* = 34, 13.5%), and gastrointestinal complaints (*n* = 16; 6.4%). There were more admissions outside the palliative care team’s working hours (*n* = 136, 54.2%). The episodes led to 91 hospitalizations (36.3%). The emergency medical or first aid ambulance was the most frequent means of admission to the emergency department (*n* = 165, 65.7%). However, there were also admissions referred by the palliative care team following consultation (*n* = 5, 2.0%) and by primary health care (*n* = 5, 2.0%). The request for an opinion from the palliative care team during an episode in the emergency department occurred in a minority of cases (*n* = 39, 15.5%). The median length of stay in the emergency department during the episode was 5 h (P25 = 3.0, P75 = 10.0 h) (Table 5).

There were eight (3.2%) admissions resulting in death in the emergency department, and the most frequent diagnosis was chronic obstructive pulmonary disease with unspecified acute exacerbation (*n* = 46, 18.3%). In four admissions (1.5%) to the emergency department, patients at admission were already in cardiorespiratory arrest, and these admissions were excluded from the analysis.

### 3.4. Associations with the Admissions to the Emergency Department

We included variables that showed association in the univariate model in a multivariable Poisson model as well as in those that showed marginal significance (*p* < 0.10). As a result, in comparison with the sign or symptom of pain, changes in behavior and status were associated with longer length of stay in the emergency department (aRR = 1.26, 95%CI = 1.06–1.50). Gastrointestinal changes (aRR = 0.77, 95%CI = 0.60–0.98), urinary tract signs or symptoms (aRR = 0.63,95%CI = 0.48–0.83), trauma (aRR = 0.59, 95%CI = 0.43–0.83), and others (aRR = 0.35, 95%CI = 0.25–0.48) were associated with a shorter length of stay in the emergency department. In comparison to access to the emergency department on one’s own initiative, an emergency medical or first aid ambulance was associated with a longer length of stay in the emergency department (aRR = 1.32, 95%CI = 1.17–1.48) (Table 6).

Patients who were most frequently hospitalized were admitted with respiratory complaints (*n* = 57, 62.6%). The in-hospital team was not asked for an opinion on palliative care in most situations—both in hospitalized (*n* = 73, 80.2%) and not hospitalized (*n* = 139, 86.9%) patients. Hospitalization during the episode in the emergency department was not associated with the variables evaluated (Table 7).

There was an association (X^2^ = 4.6, *p* = 0.032) and a higher prevalence of requests for the PC team’s opinion (*n* = 24, 20.9%) in patients admitted during service hours (Table 8).

## 4. Discussion

This study allowed us to investigate the characteristics of patients who attended a consultation to identify their profile regarding the use of health services as well as to globally analyze the episodes of admission to the emergency department.

Most patients included in this study were elderly, reflecting the important role of palliative care in the geriatric population. Worldwide, 40% of palliative care needs to address patients over 70 years of age [4]. The predictions show that the need for palliative care will increase by about 183% by 2060 [15].

The follow-up time was short and reflects late referral of patients to palliative care. Late referral collides with the approach that palliative care should be provided as early as possible in the course of the disease. Late referral is associated with poor symptomatic control, increased suffering, failure to discuss advanced care plans, caregiver burnout, and hospital deaths [16]. Earlier transition to palliative care is associated with better symptoms control, reduced distress, and more respect for patients’ preferences [17]. Early initiation of palliative care is also associated with reduced late-life acute hospital use [18].

This study enrolled all patients regardless of pathology, although oncological disease was the most frequent. Worldwide, oncological disease has the highest representation in need of intervention by palliative care, but more than 70% of needs concern other health conditions, including human immunodeficiency virus, cerebrovascular disease, and dementia. These data show the importance of including these populations in studies [13].

Most patients who use the emergency department lived at home, and most patients resorted to the emergency department despite follow-up in consultation. A greater number and time of follow-up in consultation was associated with more episodes and time spent in the emergency department. However, only a few users were advised by the palliative care team to use this service. Patients feel safer in the hospital than at home. They resorted to the emergency department due to difficulties in managing the clinical situation at home despite having caregivers [12,19]. Patients and families seek the hospital due to anxiety during episodes of worsening symptoms, lack of prior guidance by health teams, search for security and familiarity, and difficulty in accessing primary care in situations that they consider to be urgent, including outside regular office hours [20]. The existence of a home team or telephone network of health professionals with training in palliative care that is permanent and available to respond to crisis situations potentially prevents patients from going to the emergency department [20].

Among the patients who resorted to the emergency department, most died during the last admission or subsequent hospitalization. There were patients who were admitted in cardiorespiratory arrest and who died during their stay in the emergency department. Dying in hospital was associated with going to the emergency department. Patients who died during the last episode of the emergency department or subsequent hospitalization spent more time and were admitted more often into the emergency department. Patients with a higher number of emergency department admissions, hospitalizations, and hospital deaths possibly had a worse quality of life in the last months of life [21]. Care and death at home are patients’ preference [5,21]. Home care promotes greater comfort, fewer hospital infections, and cost savings [22]. Late referral of patients and, consequently, the presence of more severe diseases, might explain the difficulty in symptoms control, emergency department use, and hospital deaths [16,17].

Most hospitalizations took place following admission to the emergency department. This result may reflect the fact that the in-hospital palliative care team does not have its own inpatient unit, which leads patients and families to resort to the emergency department. Thus, there is an association between going to the emergency department and having been hospitalized at least once. Patients who had more than one hospitalization presented a greater number of admissions and a greater total length of stay in the emergency department. Patients hospitalized one or more times are twice as likely to be readmitted to the emergency department [23].

Regarding the analysis of episodes in the emergency department, the frequency of complaints matches other studies that also found pain (15–46%), respiratory disease (13–26%), and digestive symptoms (12–26%) as the main admission symptoms [7,9,12]. A wide variety of final diagnoses was obtained, but it is not statistically feasible to look for associations with the remaining variables. [7,9,12]. Gastrointestinal and genitourinary complaints were associated with a shorter length of stay in the emergency department. There was often a correlation with easily resolved clinical conditions that correspond to likely potentially avoidable episodes, such as constipation, retention, or urinary tract infection. All health professionals who have contact with palliative patients should be familiar with the management of this symptomatology to improve symptom control and them resorting to the emergency department [7,9,12].

There were more admissions after working hours. The palliative care team is only available from 8 a.m. to 5 p.m., Monday to Friday. Thus, admissions after working hours might happen because the emergency department is the only health service available in this period to respond to existing needs. [7,9]. Emergency medical or first aid ambulance was the most common means of accessing the emergency department. Health professionals involved in emergency transport recognize the advantages of death at home, but they have difficulty in promoting this deferral. The use of this resource is associated with a lack of alternatives in after-work hours, lack of clinical information about patients, and the health system’s focus on providing life support care [24].

End-of-life patients should not need to resort to an emergency department and should remain at home, or, when this is not appropriate, be directly referred to a palliative care unit. It is thus necessary to invest in training and in the organization of the network of professionals qualified to act, thereby increasing the period of availability of face-to-face, telephone, or home service. Consultancy to other professionals is also needed to support the management of crisis situations [19,25,26]. In 2019, Portugal had a ratio of 0.9 palliative medicine services per 100,000 inhabitants, which is less than the recommendations [13]. In the geographic area studied here, there is no palliative care team available to provide care at home. The existence of palliative care teams composed of specialists from different areas (specialized palliative care doctors, nurses, and family doctors) with permanent accessibility is associated with a lower frequency of admission to the emergency department. Palliative care at home significantly increases patient satisfaction, reduces the use of medical services—whether visiting the emergency department or being admitted to the hospital—and lowers end-of-life medical care costs. [27].

This study has some limitations. A retrospective analysis was performed, and hence the collection of information was dependent on its availability in clinical records. The sample reflects the reality of one region, and patients that were alive at the time of data collection were excluded, given the small number of patients. These factors should be considered for the construction and improvement of future studies on the subject. Nevertheless, this study does address a topic that lacks information both at a national and international level. The conclusions drawn here will help improve health services.

## 5. Conclusions

The aim of this study was to describe the use of the emergency department by patients followed up in palliative care consultations until death. The results show that the responses of the health system do not suit the needs of patients and families given that the majority had to resort to the emergency department. Dying in hospital was associated with going to the emergency department, and patients who died in the hospital had more admissions and spent more time there.

Among the strategies to improve the provision of care, there is the establishment and creation of action guidelines for professionals. This can combine palliative medicine services and primary care so that crisis situations can be better managed in the community. An automatic consulting system can be created for specialized clinicians based on pre-defined clinical criteria [19,28,29,30,31]. A fragmented health system cannot effectively deal with the increased demand for care due to aging populations, thus increasing long-term chronic diseases and multimorbidity. Palliative care must be integrated at all levels of health systems to ensure smooth transitions and continuity of care [32]. Undergraduate and postgraduate training in palliative care is essential to create a network of health professionals that can adequately manage the needs of patients [13]. It is necessary to invest in patient and caregiver education and establish an early care plan. There must also be organizational responses; e.g., telephone and digital assistance across more hours, for a timely response to a crisis situation. There is a need for regular contacts and communication as well as more home visits.

The use of an emergency department by palliative patients is a sign that there are gaps to be filled in the provision of care. Therefore, solutions must be implemented to respond to the growing needs in this area.

## Figures and Tables

**Table 1 ijerph-19-15204-t001:** Characterization of the patient cohort.

Characterization of the Patient Samples (*n* = 135)	*n* (%)
Sex	
Male	75 (55.6%)
Female	60 (44.4%)
Marital status	
Single	25 (18.5%)
Married	87 (64.4%)
Widow	15 (11.1%)
Divorced	8 (5.9%)
Residence	
Nursing home	13 (9.6%)
Domicile	117 (86.7%)
Host family	1 (0.7%)
Continuing care unit	4 (3.0%)
Main diagnosis	
Other causes	25 (18.5%)
Oncological disease	87 (64.4%)
Dementia syndrome	23 (17.0%)
Admissions	
0	61 (45.2%)
1	58 (43.0%)
>1	16 (11.8%)
Place of death	
Domicile	39 (28.9%)
Hospital	66 (48.9%)
Nursing Home	9 (6.7%)
Continuing care unit	20 (14.8%)
Host family	1 (0.7%)
Hospitalization proposed in consultation	
No	129 (95.6%)
Yes	6 (4.4%)
Age (years)	82.0 (67.0–88.0)
Follow-up time in consultation (months)	2.0 (1.0–5.0)
Number of consultations	5.0 (2.0–10.0)

Results presented in format *n* (%) for categorical variables and Mdn (P25–P75) for continuous variables. ED, emergency department.

**Table 2 ijerph-19-15204-t002:** Associations with admissions to the emergency department.

			Univariate Analysis	Multivariate Analysis
			*p*-Value	aOR (CI 95%)
	Did Not Go to the ED (*n* = 37)	Went to the ED (*n* = 98)		
Sex				
Male	21 (56.8%)	54 (55.1%)	Χ^2^ = 0.03, *p* = 0.863 (†)	-
Female	16 (43.2%)	44 (44.9%)		-
Marital status				
Married	21 (56.8%)	66 (67.3%)	Χ^2^ = 1.32, *p* = 0.252 (†)	-
Other	16 (43.2%)	32 (32.7%)		-
Residence				
Domicile	28 (75.7%)	89 (90.8%)	Χ^2^ = 5.33, *p* = 0.043 (‡)	1
Other	9 (24.3%)	9 (9.2%)		0.96 (0.19–4.93)
Main diagnosis				
Other causes	4 (10.8%)	21 (21.4%)	Χ^2^ = 3.23, *p* = 0.200 (†)	-
Oncological disease	24 (64.9%)	63 (64.3%)		-
Dementia syndrome	9 (24.3%)	14 (14.3%)		-
Hospitalizations				
0	30 (81.1%)	31 (31.6%)	Χ^2^ = 26.62, *p* < 0.001 (†)	1
1	6 (16.2%)	52 (53.1%)		**6.65 (1.84–24.07)** (°)
>1	1 (2.7%)	15 (15.3%)		
Death during the last visit to the ED	-	-	-	-
No	-	-	-	-
Yes	-	-	-	-
Place of death				
Domicile	20 (54.1%)	19 (19.4%)	Χ^2^ = 34.54, *p* < 0.001 (‡)	1
Hospital	4 (10.8%)	62 (63.3%)		**9.20 (2.42–35.07)**
Other	13 (35.1%)	17 (17.3%)		0.68 (0.17–2.72)
Age	84.0 (61.0–90.0)	81.0 (68.0–87.0)	U = 1601.00, *p* = 0.295 (§)	-
Follow-up time in consultation	1.0 (1.0–3.0)	3.0 (1.0–6.0)	U = 1236.50, *p* = 0.004 (§)	1.14 (0.96–1.36)
Number of consultations	3.0 (1.0–5.0)	6.0 (2.0–11.0)	U = 1307.50, *p* = 0.012 (§)	1.00 (0.91–1.10)

(†) Chi-square test; (‡) Fisher test; (§) Mann–Whitney test; aOR, adjusted odds ratios calculated via multiple logistic models including variables with *p*-value < 0.10 in the univariate analysis. Results presented in *n* format (%) for categorical variables and Mdn (P25–P75) for continuous variables. Bold text indicates statistical significance in multivariable model guided by the criterion *p*-value < 0.05 with the confidence intervals set at 95%. ED, emergency department. (°) one or more hospitalizations included in the analysis.

**Table 3 ijerph-19-15204-t003:** Associations with the number of episodes in the emergency department.

		Univariate Analysis	Multivariate Analysis
	Number	*p*-Value	aRR (CI 95%)
Sex			
Male	2.0 (1.0–4.0)	U = 986.50, *p* = 0.130 (§)	-
Female	1.5 (1.0–3.0)		-
Marital status			
Married	2.0 (1.0–3.0)	U = 882.50, *p* = 0.167 (§)	-
Other	2.0 (1.0–4.0)		-
Residence			
Domicile	2.0 (1.0–3.0)	U = 342.50, *p* = 0.461 (§)	-
Other	1.0 (1.0–3.0)		-
Main diagnosis			
Other causes	3.0 (1.0–4.0)	H = 4.11, *p* = 0.128 (¶)	-
Oncological disease	2.0 (1.0–3.0)		-
Dementia syndrome	1.5 (1.0–2.0)		-
Hospitalizations			
0	1.0 (1.0–2.0)	H = 24.52, *p* < 0.001 (¶)	1
1	2.0 (1.0–3.0)		1.11 (0.77–1.58)
>1	4.0 (3.0–7.0)		**2.26 (1.52–3.37)**
Death during the last visit to the ED			
No	1.0 (1.0–2.5)	U = 860.5, *p* = 0.043 (§)	1
Yes	2.0 (1.0–3.0)		**1.48 (1.09–2.00)**
Place of death			
Domicile	1.0 (1.0–1.0)	H = 2.34, *p* = 0.625 (¶)	-
Hospital	2.0 (1.0–3.0)		-
Other	2.0 (1.0–3.0)		-
Age	rs = −0.06	*p* = 0.528	-
Follow-up time in consultation	rs = 0.45	*p* < 0.001	**1.03 (1.01–1.06)**
Number of consultations	rs = 0.37	*p* < 0.001	1.02 (1.00–1.05)

(§) Mann–Whitney test; (¶) Kruskal–Wallis test; rs, Spearman’s correlation coefficient; aRR, adjusted relative risks calculated using multiple Poisson models, including variables with *p*-value < 0.10 in univariate analysis. Bold text indicates statistical significance in multivariable model guided by the criterion *p*-value < 0.05 with the confidence intervals set at 95%. ED, emergency department.

**Table 4 ijerph-19-15204-t004:** Associations with the total length of episodes in the emergency department.

		Univariate Analysis	Multivariate Analysis
	Length (Hours)	*p*-Value	aRR (CI 95%)
Sex			-
Male	12.5 (6.0–25.0)	U = 968.00, *p* = 0.161 (§)	-
Female	8.0 (4.0–20.0)		-
Marital status			
Married	9.0 (4.0–20.0)	U = 329.50, *p* = 0.051 (§)	1
Other	16.5 (5.0–24.0)		**1.56 (1.40–1.73)**
Residence			
Domicile	11.0 (4.0–21.0)	U = 322.00, *p* = 0.357 (§)	-
Other	7.0 (4.0–11.0)		-
Main diagnosis			
Other causes	19.0 (4.0–37.0)	H = 0.14, *p* = 0.712 (¶)	-
Oncological disease	9.0 (5.0–20.0)		-
Dementia syndrome	8.5 (4.0–18.0)		-
Hospitalizations			
0	6.0 (3.0–12.0)	H = 13.03, *p* = 0.001 (¶)	1
1	13.0 (6.0–25.0)		**1.62 (1.39–1.89)**
>1	21.0 (8.0–49.0)		**2.80 (2.37–3.32)**
Death during the last visit to the ED			
No	7.5 (3.5–17.5)	U = 864.50, *p* = 0.057 (§)	1
Yes	13.5 (6.0–24.5)		**1.78 (1.57–2.01)**
Place of death			
Domicile	7.0 (3.0–13.0)	H = 4.18, *p* = 0.124 (¶)	-
Hospital	13.5 (6.0–24.0)		-
Other	7.0 (4.0–18.0)		-
Age	rs = 0.17	*p* = 0.107	-
Follow-up time in consultation	rs = 0.40	*p* < 0.001	**1.05 (1.04–1.07)**
Number of consultations	rs = 0.36	*p* < 0.001	**1.02 (1.01–1.03)**

(§) Mann–Whitney test; (¶) Kruskal–Wallis test; rs, Spearman’s correlation coefficient; aRR, adjusted relative risks calculated using multiple Poisson models, including variables with *p*-value < 0.10 in univariate analysis. Bold text indicates statistical significance in multivariable model guided by the criterion *p*-value < 0.05 with the confidence intervals set at 95%. ED, emergency department.

**Table 5 ijerph-19-15204-t005:** Characterization of the patient sample and episodes in the emergency department.

Characterization of Episodes in the ED (*n* = 251)	*n* (%)
Main complaint	
Pain	34 (13.5%)
Respiratory	126 (50.2%)
Gastrointestinal	16 (6.4%)
Changes in behavior or general condition	34 (13.5%)
Genitourinary	14 (5.6%)
Trauma	10 (4.0%)
Other	17 (6.8%)
Outside the team’s working hours	136 (54.2%)
Hospitalization	
No	160 (63.7%)
Yes	91 (36.3%)
Origin	
Own initiative	59 (23.5%)
Ambulance	165 (65.7%)
National telephone contact center	10 (4.0%)
Consultation	5 (2.0%)
Primary health care	5 (2.0%)
Other	7 (2.8%)
Request of team’s opinion	
No	212 (84.5%)
Yes	39 (15.5%)
Time in the ED (hours)	5.0 (3.0–10.0)

Results presented in format *n* (%) for categorical variables and Mdn (P25–P75) for continuous variables. ED, emergency department.

**Table 6 ijerph-19-15204-t006:** Associations with length of stay in the emergency department.

		Univariate Analysis	Multivariate Analysis
	Length	*p*-Value	aRR (CI 95%)
Main complaint			
Pain	5.5 (3.0–9.0)	H = 25.26, *p* < 0.001 (¶)	1
Respiratory	6.0 (3.0–12.0)		1.13 (0.98–1.30)
Gastrointestinal	4.5 (2.0–9.0)		**0.77 (0.60–0.98)**
Behavior	7.0 (3.0–13.0)		**1.26 (1.06–1.50)**
Urinary	4.0 (3.0–5.0)		**0.63 (0.48–0.83)**
Trauma	3.0 (1.0–7.0)		**0.59 (0.43–0.83)**
Other	3.0 (1.0–3.0)		**0.35 (0.25–0.48)**
Number of complaints			
1	5.0 (3.0–10.0)	U = 3300.00, *p* = 0.443 (§)	-
>1	6.0 (3.0–9.0)		-
Origin			
Own	5.0 (2.0–8.0)	H = 9.54, *p* = 0.023 (¶)	1
Ambulance	6.0 (3.0–11.0)		**1.32 (1.17–1.48)**
Contact center	2.5 (1.0–4.0)		0.79 (0.58–1.08)
Other	5.0 (2.0–9.0)		0.89 (0.71–1.12)
Team’s opinion			
No	5.0 (3.0–9.0)	U = 3606.50, *p* = 0.204 (§)	-
Yes	6.0 (3.0–13.0)		-

(§) Mann–Whitney test; (¶) Kruskal–Wallis test; aRT, adjusted relative risks calculated via multiple Poisson models, including variables with *p*-value < 0.10 in univariate analysis. Bold text indicates statistical significance in multivariable model guided by the criterion *p*-value < 0.05 with the confidence intervals set at 95%. ED, emergency department.

**Table 7 ijerph-19-15204-t007:** Associations with hospitalization during an episode in the emergency department.

			Univariate Analysis	Multivariate Analysis
	Hospitalization		*p*-Value	aOR (CI 95%)
	No (*n* = 160)	Yes (*n* = 91)		
Main complaint				
Pain	25 (15.6%)	9 (9.9%)	Χ^2^ = 11.02, *p* = 0.088 (†)	1
Respiratory	69 (43.1%)	57 (62.6%)		2.21 (0.94–5.16)
Gastrointestinal	11 (6.9%)	5 (5.5%)		1.27 (0.34–4.77)
Behavior	22 (13.8%)	12 (13.2%)		1.49 (0.52–4.29)
Urinary	11 (6.9%)	3 (3.3%)		0.71 (0.16–3.19)
Trauma	8 (5.0%)	2 (2.2%)		0.72 (0.12–4.11)
Other	14 (8.8%)	3 (3.3%)		0.58 (0.13–2.54)
Number of complaints				
1	138 (86.3%)	80 (87.9%)	Χ^2^ = 0.14, *p* = 0.708 (†)	-
>1	22 (13.8%)	11 (12.1%)		-
Origin				
Own	42 (26.3%)	17 (18.7%)	Χ^2^ = 6.50, *p* = 0.090 (†)	1
Ambulance	97 (60.6%)	68 (74.7%)		1.66 (0.86–3.21)
Contact center	9 (5.6%)	1 (1.1%)		0.27 (0.03–2.35)
Other	12 (7.5%)	5 (5.5%)		1.05 (0.31–3.54)
Team’s opinion				
No	139 (86.9%)	73 (80.2%)	Χ^2^ = 1.96, *p* = 0.162 (†)	-
Yes	21 (13.1%)	18 (19.8%)		-

Results are presented in format n (%) for categorical variables and Mdn (P25–P75) for continuous variables; (†) chi-square test; aOR, adjusted odds ratios calculated via multiple logistic models including variables with *p*-value < 0.10 in the univariate analysis.

**Table 8 ijerph-19-15204-t008:** Associations with time of admission to the emergency department, including the palliative care team’s work schedule.

			Univariate Analysis	Multivariate Analysis
	Time of Admission		*p*-Value	aOR (CI 95%)
	Team Available (*n* = 115)	Team Unavailable (*n* = 136)		
Main complaint				
Pain	13 (11.3%)	21 (15.4%)	Χ^2^ = 4.43, *p* = 0.619 (†)	-
Respiratory	61 (53.0%)	65 (47.8%)		-
Gastrointestinal	8 (7.0%)	8 (5.9%)		-
Behavior	18 (15.7%)	16 (11.8%)		-
Urinary	5 (4.3%)	9 (6.6%)		-
Trauma	5 (4.3%)	5 (3.7%)		-
Other	5 (4.3%)	12 (8.8%)		-
Number of complaints				
1	101 (87.8%)	117 (86.0%)	Χ^2^ = 0.18, *p* = 0.675 (†)	-
>1	14 (12.2%)	19 (14.0%)		-
Origin				
Own	27 (23.5%)	32 (23.5%)	Χ^2^ = 0.49, *p* = 0.920 (†)	-
Ambulance	75 (65.2%)	90 (66.2%)		-
Contact center	4 (3.5%)	6 (4.4%)		-
Other	9 (7.8%)	8 (5.9%)		-
Team’s opinion				
No	91 (79.1%)	121 (89.0%)	**Χ^2^ = 4.60, *p* = 0.032 (†)**	-
Yes	24 (20.9%)	15 (11.0%)		-

Results presented in format n (%) for categorical variables and Mdn (P25–P75) for continuous variables; (†) chi-square test; aOR, adjusted odds ratios calculated via multiple logistic models including variables with a *p*-value < 0.10 in the univariate analysis. Bold text indicates statistical significance in univariate analysis guided by the criterion *p*-value < 0.05.

## Data Availability

The data supporting the reported results can be found in the Póvoa de Varzim/Vila do Conde Hospital Center archive.

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
