# Peer review of "Admission to the Emergency Department by Patients Being Followed up for Palliative Care Consultations"

_ijerph, 2022, doi:10.3390/ijerph192215204_

Round 1
Reviewer 1 Report
The findings of the study are very relevant in the context of palliative care for improving the quality of life of will persons and their families, in Portugal. I consider that the article has conditions to be published, however, the authors must correct the following aspects:
Line 65 – remove the curved parenthesis in the expression “random sampling [14)].”
Line 139 – put the word “(Tables 4)” in the singular.
Author Response
Thank you for all the suggestions that contributed to the improvement of our paper.
We removed the curved parenthesis in line 70 pag 2 and corrected the word “Tables” to “Table” (line 158 pag 6).
The article was submitted to an English editing service prior submission for publication in this journal, if needed we can send the certification of the English edition.
Reviewer 2 Report
My major concerns relate to the inadequate and unclear description of the palliative care intervention and the nature of palliative care teams. If I am correct that the study simply tracks folks seen during a hospitalization with no community based palliative care access or engagement, nor any hospice support, the snapshot of their high rate of re-presentation is entirely expected and does not add to or illuminate the phenomena highlighted. If the behavior of this cohort was compared to case-matched controls not receiving the palliative service, more relevant findings may have surfaced.
That being said, more clear description of the palliative care treatment model that defines the cohort, including what may have been missing in the community, might simply profile the cohort's behavior using the ED and hospital in the context of similar communities with similar access to limited services.
Author Response
Thank you for the suggestions that contributed to the improvement of our paper.
We agree that the methodology can be improved in future studies to include a case matched control. This is the first study carried out in our country related to emergency department use by palliative care patients and we consider this study the starting point.
We included information about the working model of the palliative care team in our region “ The hospital center where the study took place has a palliative care team whose mission is to support patients and families in need of differentiated palliative care. It is a multidisciplinary team which includes doctors, nurses, a psychologist and a social worker, working Monday to Friday, 8 am to 5pm. Referral to the consultation can be carried out by doctors of any of the hospital services or by family doctors. The team has a telephone contact, available Monday to Friday, from 8am to 5pm, for patients and families and also for requests of advice from other health professionals. It is a consulting team without its own inpatient service and does not provide home care services. Home palliative care service is not available in this region.” (line 76-84 pag 2).
Reviewer 3 Report
Thank you for inviting me to review this paper. This was an interesting topic in an important area.
The introduction was written well, but the overall study rationale was not described in much detail and needs to be stronger. A large number of analyses are conducted; to include this number of analyses a stronger justification for each and/or clear hypotheses are needed.
The power analysis is described in lines 62-65 but this needs elaborating. What was the “necessary sample size” identified from this analysis?
Why is the analysis limited to only patients who had died? The introduction describes the use of palliative care in instances of chronic conditions and not just end of life. The authors need to explain their reasoning for then limiting this analysis in this way. Similarly, since palliative care consultations were done in 2019, the sample is limited to patients who died relatively soon after this consultation (within a couple of years) and could this have excluded patients who were less acutely ill?
Following from this, it’s not surprising that ED visits were linked to hospitalisations, since your sample is skewed towards sicker people, many of whom died after being admitted to the ED. Could it not be argued that these patients were clearly in need of intensive medical care and would have needed hospitalisation even if a palliative consultation service had been available at that time?
For clarity, is the “number of consultations” in the table referring to the number of palliative consultations? Is this between the first 2019 consultation and death?
Table 2 – Hospitalisations has three levels and there should be two odds ratios reported in the Multivariate column.
Page 10, line 260-62 needs a reference for this statement.
Overall, this was an interesting study idea and analysis. The paper needs a stronger central argument to pull this together.
Author Response
Thank you for all the suggestions that contributed to the improvement of our paper.
- We completed our objectives to justify the number of analyses we conducted as you kindly suggested “This study aimed to characterize the use of the emergency department by patients followed up in palliative care consultations between time of admission and time of death. Firstly we wanted to characterize and identify differences between users and non-users of the emergency department. Secondly we intended to characterize patients and the episodes in the emergency department and identify factors associated with the use of emergency department.” (lines 58-63 pag 2).
- We corrected the sentence to give more information about the sample size “We considered an average of 207 per year and sample size was calculated taking into account a confidence level of 95% and a margin error of 5% with random sampling and obtained a necessary sample of 135 patients.” (lines 66-68 pag 2).
- The analysis is limited to patients who had died since that our objective was to analyze the use of emergency department of the patients between the admission to palliative care consultation and death. There was a total of 207 patients that had their first consultation in the year of 2019 and 10 were still alive at the time of data collection at the end of May 2022. It was a small number and we decided to exclude them, since we wouldn’t be able to compare samples. The outcome death was considered. It is true that the patients who were excluded could have less severe pathologies and it might be a limitation of our study. We completed the sentence to justify our decision to exclude them “The study included deceased adults aged 18 years or more on whom there was complete information in the clinical files. Patients who were alive at the time of data collection, in May 2022, were excluded (n=10).” (lines 70-74 pag 2) and we included that information when we describe the limitations of our study “This study does have some limitations. A retrospective analysis was performed, and hence the collection of information was dependent on its availability in clinical records. The sample reflects the reality of one region and patients that were alive at the time of data collection were excluded, given the small number of patients. These factors should be considered for the construction and improvement of future studies on the subject” (line 304-308 pag 11).
- Our patients died soon after admission to palliative care consultations and we completely agree that they might have more severe conditions. One of the main explanations is that they are being referred late to the palliative consultation and it is also an important conclusion of our study since the late referral to palliative care is a big concern that must be addressed for efficient and adequate palliative care, both in this region as well as at a worldwide level (lines 224-230 pag 10). We also included information about the palliative care team working objectives and how the referral process takes place to clarify the limitations of the palliative care service of our region that also explain the need of the emergency department use “The hospital center where the study took place has a palliative care team whose mission is to support patients and families in need of differentiated palliative care. It is a multidisciplinary team which includes doctors, nurses, a psychologist and a social worker, working Monday to Friday, 8 am to 5pm. Referral to the consultation can be carried out by doctors of any of the hospital services or by family doctors that work in the region. The team has a telephone contact, available Monday to Friday, from 8am to 5pm, for patients and families and also for requests of advice from other health professionals. It is a consulting team without its own inpatient service and does not provide home care services. Home palliative care service is not available in this region.”(lines 76-84 pag 2). We also included the following sentence “Late referral of patients and, consequently, the presence of more severe diseases, might explain worse symptoms control, emergency department use and hospital deaths [16,17].” (line 258-260, pag 10).
- The number of consultations in the table is between the first 2019 palliative care consultations and death. We completed the sentence to improve the understanding of the variables – “Data were collected to characterize the patients and their use of health services: gender, age, marital status, residence (domicile, nursing home, continuing care unit, host family), time between admission to palliative care consultation and death, number of consultations between admission to palliative care consultation and death, number of hospitalizations between admission to palliative care consultation and death, time in the emergency department, place of death, death in the last admission to the emergency department or subsequent hospitalization, and hospitalization proposed in palliative care consultation” (lines 86-91 pag 2).
- In table 2 we included a note explaining having only 2 level for the analysis related to hospitalizations, we considered non-hospitalization and one or more hospitalizations. We included the note “(Ëš) one or more hospitalizations included in the analysis.”
- We corrected the sentence to be in line with the citations “There were more admissions after working hours. The palliative care team is only available from 8am to 5pm, Monday to Friday. Thus, admissions after working hours might happen because the emergency department is the only health service available in this period to respond to existing needs [7,9].” (line 280-283 pag 11).